The association between fibroblast growth factor 21 with diabetes retinopathy among type 2 diabetes mellitus patients: a systematic review, meta-analysis, and meta-regression

Basir Herni 1 hernibasir@yahoo.co.id
http://orcid.org/0000-0002-5450-4568 Nugrahani Annisa Salsabilla Dwi 2
Aman Andi Makbul 1
Bakri Syakib 3
Rasyid Haerani 3
Umar Husaini 1
H. P. Faridin 3
Ichsan Andi Muhammad 4
Zainuddin Andi Alfian 5
1 Endocrinology and Metabolism Division, Department of Internal Medicine, Faculty of Medicine, Hasanuddin University , Makassar , Indonesia
2 Faculty of Medicine, Universitas Airlangga , Surabaya, East Java , Indonesia
3 Department of Internal Medicine, Faculty of Medicine Hasanuddin University , Makassar , Indonesia
4 Department of Ophthalmology, Faculty of Medicine, Hasanuddin University , Makassar , Indonesia
5 Department of Public Health, Faculty of Medicine, Hasanuddin University , Makassar , Indonesia
Suner Aslı
Electronic publication date: 2024 Dec 13
Publication date: 2024
Volume: 12
Electronic Location ID: e18308
Received 2024 Jul 22; Accepted 2024 Sep 24
Copyright: © 2024 Basir et al.
Copyright year: 2024
Copyright holder: Basir et al.
License: This is an open access article distributed under the terms of the Creative Commons Attribution License, which permits unrestricted use, distribution, reproduction and adaptation in any medium and for any purpose provided that it is properly attributed. For attribution, the original author(s), title, publication source (PeerJ) and either DOI or URL of the article must be cited.
License URL: https://creativecommons.org/licenses/by/4.0/

Keywords: Diabetic retinopathy, Fibroblast growth factor, Type 2 diabetes mellitus, Meta-analysis, Meta-regression

Funding: The authors received no funding for this work.

==============================
Background

Diabetic retinopathy (DR), a leading cause of vision loss worldwide, is a common complication of type 2 diabetes mellitus (T2DM) driven by chronic hyperglycemia and microvascular damage. Fibroblast growth factor 21 (FGF21) is crucial in blood sugar regulation and has been linked to DR incidence and severity. While some studies suggest that FGF21 levels may contribute to the DR incidence, others propose a protective role. This discrepancy necessitates further analysis, prompting this study to evaluate the association between FGF21 levels and DR incidence and severity in T2DM patients.

Methods

A systematic search was conducted through MEDLINE, Web of Science, Scopus, and Embase up to May 2024 for studies evaluating the association between FGF21 and DR incidence and severity. A random-effect model meta-analysis was performed to calculate the pooled standardized mean difference (SMD) and 95% confidence intervals (CI). A univariate meta-regression was performed to analyze factors influencing pooled size estimates. All statistical analyses were performed using STATA 17 software.

Result

This systematic review and meta-analysis of 5,852 participants revealed that FGF21 was positively correlated with DR (SMD 3.11; 95% CI [0.92–5.30], p = 0.005) and sight-threatening DR (STDR) incidence (SMD 3.61; 95% CI [0.82–6.41], p = 0.01). There was no significant difference in FGF21 levels in DR vs STDR (p = 0.79). Subgroup analysis revealed a significant difference in DR incidence between LDL groups, with higher DR incidence in the group with low-density lipoprotein (LDL) levels >100 (P < 0.00001). Meta-regression revealed no variables significantly influenced the pooled size estimates.

Conclusion

A higher level of FGF21 was associated with higher DR and STDR incidence among T2DM patients, highlighting its potential utilization as a biomarker for DR detection and enabling the exploration of FGF21-based treatment strategies. However, variables independently predicting DR among patients with elevated FGF21 levels shall be explored further.

PROSPERO ID

CRD42024559142.

Introduction

Diabetes mellitus (DM) is a chronic metabolic disorder characterized by elevated blood sugar levels arising from defects in insulin secretion, insulin action, or both (American Diabetes Association, 2014; Ye et al., 2023). Prolonged elevation of blood sugar levels can lead to damage and dysfunction of several specific organs, such as the eyes, kidneys, nerves, heart, and blood vessels. Various epidemiological studies indicate a tendency towards increased incidence and prevalence of type 2 diabetes mellitus (T2DM) worldwide (American Diabetes Association, 2021). In 2019, the estimated number of adults aged 20 to 79 years suffering from DM was 463 million, accounting for 9.3% of the total adult population globally (Saeedi et al., 2019). This number is projected to reach 578 million, or approximately 10.2% of the total adult population worldwide in 2030 (Saeedi et al., 2019).

T2DM is known as the most prevalent type of DM, accounting for over 90% of all diabetes cases worldwide. Characterized by a progressive loss of insulin secretion due to insulin resistance, T2DM is known as the leading cause of death and disability worldwide, mainly due to diabetes-related complications. Complications of DM can manifest as vascular disturbances in both microvascular and macrovascular systems and dysfunction in the nervous system or neuropathy (American Diabetes Association, 2014).

Macrovascular complications, such as those affecting the heart, brain, and blood vessels, are significant concerns in diabetes management. However, microvascular complications, particularly (Saeedi et al., 2019; Sinclair & Schwartz, 2019) diabetic retinopathy (DR), pose a substantial burden among the working-age population worldwide. Not only that, it contributes as the leading cause of visual impairment worldwide; 50–60% of T2DM patients are estimated to experience DR, and 2.6% of them have vision loss (American Diabetes Association, 2021; Wang & Lo, 2018).

DR is broadly classified into two categories: nonproliferative diabetic retinopathy, described as DR, and advanced proliferative or sight-threatening DR (PDR/STDR) (Duh, Sun & Stitt, 2017). The two categories were differentiated by the presence of pathologic retinal neovascularization, which is known to be a hallmark of STDR. This pathological condition arises from chronic hyperglycemia-induced vascular damage, mediated through the activation of the polyol pathway, the accumulation of advanced glycation end products (AGEs), the protein kinase C (PKC) pathway and the hexosamine pathway, all of which contribute to oxidative stress (Brownlee, 2005). These processes lead to the loss of pericyte, a defining hallmark of DR. The loss of pericytes, which provides structural support, results in capillary walls outpouching, further contributing to microaneurysm formation (Wang & Lo, 2018).

As these processes advance, hemorrhages from this newly formed aneurysm may present and impair vision (Wang & Lo, 2018). This condition can severely decrease the quality of life and productivity, especially among working-age groups. Notably, among those who experienced DR, early-stage DR is often asymptomatic, resulting in delayed diagnosis and treatment initiation. This delay contributes to the high prevalence of undiagnosed DR and sight-threatening DR (STDR), underscoring the urgent need for comprehensive eye health services, including regular vision screening for individuals with DM (Frudd et al., 2022).

In addition to traditional biomarkers such as HbA1c, microalbuminuria, and urine albumin creatinine ratio (UACR), various molecular biomarker assays have emerged for predicting and assessing the incidence of DR. These biomarkers are associated with mechanisms involved in the occurrence of DR such as hypoxia, oxidative stress, inflammation, endothelial dysfunction, and angiogenesis (Sinclair & Schwartz, 2019; Praidou et al., 2011; Frudd et al., 2022).

Human fibroblast growth factor (FGF) comprises 22 groups generally divided into three subfamilies: paracrine, endocrine, and non-signal FGF (Kharitonenkov et al., 2005). Fibroblast growth factor 21 (FGF21), a member of the endocrine subfamily, is produced in the liver and is a 210-amino acid polypeptide that plays a crucial role in atherosclerosis, blood sugar regulation, and lipid metabolism (Jin, Xia & Han, 2021; Szczepańska & Gietka-Czernel, 2022).

FGF21 is expressed in response to stress triggers like oxidative stress from reactive oxygen species (ROS), and it has been found to interact with a high-affinity receptor called β-klotho, which functions as a single-pass transmembrane protein (Ogawa et al., 2007). Interestingly, FGF21 has complex relationships with T2DM, which initially plays a protective role in the early phase of DM by improving glucose homeostasis through reducing β-cells apoptosis and dysfunction via PPARδ/γ signaling pathways (Xie et al., 2019). However, increased concentrations of FGF21 are paradoxically observed in heightened insulin resistance, obesity, diabetes, and metabolic syndrome. Despite its unclear paradoxical increase mechanisms, this phenomenon is suggested as the result of resistance and compensatory response to a dysregulated metabolic state (Woo et al., 2013; Cheung & Deng, 2014; Jung et al., 2017).

Despite several studies that reported a significant correlation between increased FGF21 levels and the incidence of DR (Jin, Xia & Han, 2021; Lin et al., 2014; Wang et al., 2019), another study revealed no significant association between FGF21 levels and risk of DR incidence within 5 years (Ong et al., 2015). Moreover, another study demonstrated that FGF21 levels are not significantly associated with DR incidence (Mousavi et al., 2017). Hence, these controversial conclusions regarding the significance of FGF21 as a predicting biomarker for DR’s incidence and severity warrant further analysis. In addition, comprehensive analysis of independent factors predicting DR incidence remains varied. Therefore, this study aims to analyze the association between serum FGF21 levels and the incidence and severity of DR in T2DM patients. Understanding the association between FGF21 levels will potentially enable its use as a biomarker for DR detection and allow targeted therapeutic strategies in T2DM and DR management.

Materials and Methods

This systematic review and meta-analysis was conducted per the Preferred Reporting Items for Systematic Reviews and Meta-Analyses (PRISMA) 2020 guidelines (Higgins et al., 2021). This review has been registered on PROSPERO under the number CRD42024559142. The protocol is described as follows.

Search strategy

A systematic search was conducted on MEDLINE, Web of Science, Scopus, and EMBASE with coverage up to April 2024 was performed initially using the following keywords: “fibroblast growth factor 21” AND “diabetic retinopathy,” and their synonyms, which combined using Boolean operators. Complete details of the search strategy are available in Material S2. The search results from all databases were exported and imported into Covidence, software for literature screening in systematic reviews. All titles and abstracts from the search were cross-referenced to identify duplicates and any potential missing studies. Titles and abstracts were screened for a subsequent full-text review. Two authors (ASDN and HB) independently performed the complete search strategy. Any disagreements were resolved through discussion with the referee or third author (MA).

Inclusion and exclusion criteria

The inclusion criteria for studies were: (1) written in English; (2) evaluating the association of fibroblast growth factor 21 with DR incidence and severity; (3) observational study design (cross-sectional study, cohort, or case-control); (4) measured and reported the association between FGF21 levels and DR incidence/severity in numerical values; and (5) human subjects. Exclusion criteria included: (1) duplicate reports; (2) no full-text available; (3) conference abstracts, review, case reports, case series, and meta-analysis; and (4) studies with insufficient data to extract.

Study selection and data extraction

After a review of abstracts, relevant articles were retrieved and reviewed for further analysis. Bibliographies of these articles provided further references. Two independent reviewers (ASDN and HB) reviewed all retrieved records. Uncertainties were resolved via discussion with a third reviewer (MA). Data extracted from each included study were first author’s name, publication year, study design, country, sample size, age, sex proportion, duration of diabetes, the levels of FGF21 in T2DM patients without DR, with DR, and sight-threatening DR (STDR).

Data extraction and quality assessment

Relevant articles were thoroughly identified, and their information, such as author, publication year, country, study design, sample size, age, FGF21 levels, BMI, and HbA1c, were extracted. Two reviewers will independently assess the risk of bias in included studies using the Newcastle–Ottawa Scale (NOS) for observational studies (cohort, case-control, and cross-sectional studies) to assess participant selection, comparability between groups, and ascertainment of exposure or outcome. Ratings ranged from zero to nine, categorizing studies as poor quality (zero), fair quality (three to five), and excellent or high quality (six to nine). A third author was brought in in conflicting assessments to help reach an agreement (Stang, 2010).

Statistical analysis

Meta-analysis was conducted to estimate the pooled effect size of FGF21 levels and its association with T2DM and DR. Descriptive data of the participants’ characteristics are reported as mean ± standard deviation (SD). Descriptive analyses and figures of risk of bias were performed using a spreadsheet (Microsoft Excel 2016, Microsoft, Redmond, WA, USA). In contrast, meta-analytic statistics were calculated using STATA version 17 (Stata Corporation, College Station, TX, USA). The standardized mean difference (SMD), the number of participants, and the standard error of the SMD for each study were used to quantify changes in the performance variables when comparing the level of FGF21 in patients with T2DM without DR, with DR, and with STDR. SMDs for each study group were calculated using Hedges’s g (Brydges, 2019). SMDs were weighted by the inverse of variance to calculate an overall effect and its 95% confidence interval (CI). The net treatment effect was obtained by subtracting the SMD of the control group from the SMD of the experimental group. The variance was calculated from both groups’ pooled SD of change scores. Subgroup analysis was performed for race, high-density lipoprotein (HDL), low-density lipoprotein (LDL), total cholesterol (TC), and triglycerides (TG) levels.

A univariate, random-models meta-regression analysis was performed to investigate whether clinical or laboratory indices could independently predict DR incidence in the pooled analysis. The independent variables examined included HbA1c, age, sex, race, duration of DM, systolic blood pressure (SBP), LDL cholesterol, HDL cholesterol, TG, and TC levels. For the dependent variable (DR incidence), the effect of each independent variable was tested. The restricted maximum likelihood method is employed under the random effects model. A significant p-value was defined as less than 0.1. The tau2 and I2 statistical indices were used to assess heterogeneity.

To avoid problems using Q statistics to assess heterogeneity, I2 statistics was calculated, indicating the percentage of observed total variation across studies due to absolute heterogeneity rather than chance. I2 interpretation is intuitive and lies between 0% and 100%. An I2 value between 25% and 50% represents a small amount of inconsistency, an I2 value between 50% and 75% represents a medium amount of heterogeneity, and an I2 value >75% represents a large amount of heterogeneity. A restrictive categorization of values for I2 would not be appropriate for all circumstances, although it would tentatively accept adjectives of low, moderate, and high to I2 values of 25%, 50%, and 75%, respectively (Higgins et al., 2003).

Sensitivity analysis

Sensitivity analysis was used to determine whether any single study or group of studies significantly influenced the overall results. The leave-one-out method eliminated a single study at a time. If substantial heterogeneity occurred, subgroup analysis was employed to find the sources of heterogeneity. The leave-one-out method omits one study at a time and was performed using STATA version 17.

Publication bias

Publication bias and small-study effects were assessed visually using a funnel plot and statistically using Egger’s test (Sterne & Egger, 2001). The asymmetrical or disproportional distribution data in the funnel plot evidenced the presence of publication bias. In contrast, the absence of publication bias was suggested when the data were distributed approximately symmetrically. Additionally, a significant p-value of Egger’s test indicates the presence of publication bias and small study effects (Egger et al., 1997). Egger’s test was calculated using STATA version 17.

Results

Study selection

A total of 248 studies were initially obtained from five databases (MEDLINE, Web of Sciences, EMBASE, ScienceDirect, and Scopus) and manually from the references of included studies. Among them, 127 duplicate records were removed automatically before screening. During the screening process, 83 articles with irrelevant titles/abstracts were excluded, leaving 38 potential ones for further identification. A total of 31 studies were excluded due to unsuitable study design (review/case report/letters to the editor), including other type(s) of diabetes or complication(s), different fibroblast growth factors, irrelevant outcome(s), or unavailable full-text. Seven studies fulfilled the criteria and were then assessed for study quality, and all of them were included in the pooled analysis. The detail of the study flow diagram (PRISMA) can be seen in Fig. 1.

Figure 1 PRISMA flow diagram of the included studies.

Characteristic of included study

Eventually, seven studies were included for qualitative synthesis (Heidari & Hasanpour, 2021; Mousavi et al., 2017; Jung et al., 2017; Jin, Xia & Han, 2021; Esteghamati et al., 2016; Lee et al., 2023; Lin et al., 2014) (Table 1), and six studies incorporating 5,852 participants (710 NDR, 356 with NPDR, and 4,786 with STDR) were finally pooled in a meta-analysis. The six included studies published between 2014 and 2021, with the sample size ranging from 47 to 4,760. Three studies were conducted in Iran, two in China, and one in South Korea. Five studies compared the mean FGF21 levels in DR and NDR and four compared the mean FGF21 levels in NDR and STDR. Additionally, the levels of serum FGF21 were compared in three included studies. The characteristics of the included study are detailed in Table 1.

Table 1 Characteristics of included study.

Author	Year	Country	Study design	Sample size
NDR/DR/STDR	FGF-21 cut off value	NOS	
Esteghamati et al. (2016)	2016	Iran	Cross-sectional	44/46	233.00 (109.00) pg/mL in NPDR and 215.00 (122.00) pg/mL in STDR, (p = 0.361).	8	
Heidari & Hasanpour (2021)	2021	Iran	Cross-sectional	93/44/47	DR prediction with FGF-21 >312 pg/ml, with sensitivity of 97.80% and specificity of 96.77%.	9	
Jung et al. (2017)	2016	South Korea	Cross-sectional	227	OR for the DR incidence was 0.08 for the FGF21 second tertile when compared with the first tertile (p = 0.029). OR of retinopathy in third tertile group was lower than first tertile and higher than second tertile, but statistically insignificant.	7	
Jin, Xia & Han (2021)	2021	China	Cross-sectional	345/207/102	Serum FGF21 level was noted as an independent risk factor for DR and STDR (p < 0.01). Serum FGF21 level >478.76 pg/mL suggested the occurrence of DR and that level >554.69 pg/mL indicated STDR (p < 0.01).	8	
Mousavi et al. (2017)	2017	Iran	Cross-sectional	22/25	Serum FGF-21 predicts DR with the cutoff of 196 pg/mL, with a sensitivity of 80% and specificity of 47.2%.	7	
Lin et al. (2014)	2014	China	Cross-sectional	34/34/49	The estimated cut-off value of FGF21 is 550 pg/mL, with 86.5% sensitivity and 75% specificity for the existence of diabetic retinopathy (area under the curve = 0.776, p > 0.05).	8	
Lee et al. (2023)	2023	China	Retrospective cohort	4,760	FGF-21 did not significantly predict DR incidence (HR 1.10 (0.96–1.26), p = 0.16)	9	
Note:

DR, diabetic retinopathy; NPR, non-proliferative diabetic retinopathy; STDR, sight-threatening diabetic retinopathy.

Study quality assessment

The quality assessment of each included study was performed using NOS. All six included studies were rated moderate to high quality. The quality assessment of each study using the NOS critical appraisal checklist is listed in Table 1.

Association between FGF21 and DR incidence

Five studies involving 889 patients reported serum FGF21 levels for DR. The present study demonstrates a significant positive association between FGF21 levels and DR incidence (SMD 3.11; 95% CI [0.92–5.30], p = 0.005) (Fig. 2). This indicates that higher levels of FGF21 predict the incidence of DR among T2DM patients.

Figure 2 Association between FGF-21 levels with DR incidence (Jin, Xia & Han, 2021; Lin et al., 2014; Esteghamati et al., 2016; Heidari & Hasanpour, 2021; Mousavi et al., 2017).

Association between FGF21 with STDR incidence

Further analysis was performed to calculate the association between FGF21 in T2DM patients without DR and T2DM with STDR. Four pooled studies found a significant positive association between FGF21 levels and STDR (SMD 3.61; 95% CI [0.82 to 6.41], p = 0.01) (Fig. 3). Significant heterogeneity was found in this pooled analysis. A larger effect size was found in STDR than in DR incidence, indicating a possible association between FGF21 levels and DR severity.

Figure 3 Association between FGF-21 levels with STDR incidence (Jin, Xia & Han, 2021; Lin et al., 2014; Heidari & Hasanpour, 2021; Mousavi et al., 2017).

Association between FGF21 levels with DR severity

Five studies were included in this effect size estimate to evaluate the association between DR severity and serum FGF21 levels. However, it was revealed that the association between serum FGF21 levels and DR severity among NPDR and STDR patients was insignificant (p = 0.79) (Fig. 4).

Figure 4 Association between FGF-21 levels with DR severity (Jin, Xia & Han, 2021; Lin et al., 2014; Esteghamati et al., 2016; Heidari & Hasanpour, 2021; Mousavi et al., 2017).

Subgroup analysis

Subgroup analyses investigating the association between serum FGF21 levels and DR incidence are depicted in Fig. 5. From the subgroup analysis, larger pooled effect sizes were observed among Asian populations (SMD 4.71; 95% CI [1.33–8.10], p = 0.006), individuals with higher HDL levels (>40 mg/dL) (SMD 4.09; 95% CI [1.12–7.07], p = 0.007), higher LDL levels (>100 mg/dL) (SMD 6.50; 95% CI [5.28–7.71], p < 0.001), and lower TG levels (<130 mg/dL) (SMD 3.78; 95% CI [2.26–5.31], p < 0.001).

Figure 5 Summary of subgroup analysis.

However, significant differences across all subgroups were only evident in the pooled analysis of LDL levels, where LDL levels >100 mg/dL were significantly associated with higher DR incidence in the pooled samples (SMD 6.50; 95% CI [5.28–7.71], p < 0.001). In contrast, no statistically significant differences were observed among subgroups defined by race, HDL, triglyceride (TG), and total cholesterol (TC) levels, as indicated by non-significant tests of subgroup difference.

Sensitivity analysis

Our sensitivity analysis using the leave-one-out method found no significant changes in the pooled estimates of Hedges’s g when excluding each study one at a time (Fig. 6). Omitting any study from the pooled analysis did not affect the statistical significance of the overall outcomes. This finding indicates that no single study substantially impacted the overall findings.

Figure 6 Sensitivity analysis (Jin, Xia & Han, 2021; Lin et al., 2014; Esteghamati et al., 2016; Heidari & Hasanpour, 2021; Mousavi et al., 2017).

Publication bias

The funnel plot analysis suggested no obvious evidence of publication bias in the pooled estimates, as the plot displayed approximately symmetrical (Fig. 7). Regression-based Egger’s test analysis revealed insignificant estimates, indicating no small study effects in the pooled analysis (Z = −1.54, p = 0.12).

Figure 7 Funnel plot.

Meta-regression

Univariable meta-regression analysis investigated whether clinical or laboratory indices could independently predict DR incidence in T2DM patients. However, none of the examined clinical indices (including HbA1c, age, sex, race, duration of DM, systolic blood pressure (SBP), LDL cholesterol, HDL cholesterol, and TC levels) were found to be significant independent predictors of DR incidence in this population. Detailed results of the meta-regression are summarized in Table 2.

Table 2 Summary of meta-regression.

Variate	Estimate, 95% CI	p-value	
HbA1c	0.03 [−4.65 to 4.72]	0.988	
Age	0.78 [−0.07 to 1.64]	0.075	
Sex	0.09 [10.1–0.28]	0.348	
Race	−2.65 [−6.82 to 1.50]	0.210	
Duration of DM	−0.604 [−7.9 to 6.69]	0.871	
SBP	0.20 [−1.89 to 2.29]	0.851	
LDL	−4.19 [−8.51 to 0.11]	0.056	
HDL	2.42 [−1.90 to 6.74]	0.273	
TG	−1.16 [−6.15 to 3.89]	0.648	
TC	1.21 [−3.75 to 6.18]	0.633	

Discussion

To our knowledge, this study is the first systematic review and meta-analysis to analyze the association of the protein FGF21 with DR. Our study demonstrates that higher FGF21 levels predict the incidence of DR and STDR among T2DM patients. This study provides robust data to further validate the utilization of FGF21 as a biomarker of DR in T2DM patients.

In chronic hyperglycemia, there is an accumulation of advanced glycation end products (AGEs), activation of protein kinase C (PKC), dysregulation of polyol pathways, and activation of hexosamine pathways. All these factors trigger oxidative stress, leading to basement membrane thickening, retinal ischemia, increased vascular endothelial growth factor (VEGF), and neovascularization, which cause non-proliferative and proliferative DR (Duh, Sun & Stitt, 2017). In other pathways, pericyte damage, endothelial dysfunction, blood-retinal barrier (BRB) damage, and increased vascular permeability lead to macular edema (Frudd et al., 2022; Ansari et al., 2022).

FGF21 is an endocrine hormone primarily synthesized by the liver and adipose tissue, regulated by peroxisome proliferator-activated receptors (PPAR)δ and PPARγ, that plays a role in glucose metabolism, fat, insulin resistance, and obesity (Gómez-Sámano et al., 2017; Fisher & Maratos-Flier, 2016; Ornitz & Itoh, 2015; Tomita et al., 2020b). Conditions associated with increased oxidative stress, such as hyperglycemia, lead to elevated levels of FGF21, which has complex effects on T2DM (Gómez-Sámano et al., 2017; Iacobini et al., 2021). Initially, elevated FGF21 levels protect DM by improving glucose metabolism through several liver-mediated pathways. Firstly, FGF21 reduces insulin resistance by stimulating insulin secretion via the PI3K/Akt signaling pathway, enhancing postprandial insulin sensitivity (Rusu et al., 2017; Tan et al., 2023). Secondly, FGF21 protects pancreatic β-cells by promoting islet autophagy, which is mediated by the activation of AMPK-acetyl coenzyme A carboxylase (ACC) and PPARδ/γ signaling pathways, which contributes to the survival and functionality of β-cell, preventing their dysfunction (Erickson & Moreau, 2017). Thirdly, FGF21 enhances insulin sensitivity by inhibiting hepatic mTORC1, further elucidating its protective role against DM (Rusu et al., 2017; Erickson & Moreau, 2017; Geng, Lam & Xu, 2020). However, the protective effects of FGF21 appear to be most significant in the early stages of hyperglycemia, as FGF21 appears to be elevated in chronic hyperglycemia as a compensatory or resistance effect (Jung et al., 2017).

Interestingly, our study found an association between increased FGF21 levels and the incidence and severity of DR. Although the mechanisms linking serum FGF21 levels to DR are poorly understood, it is hypothesized that elevated FGF21 in diabetic complications emerges due to ‘FGF21 resistance’ or a paradoxical increase (Yang et al., 2012). As previously noted in animal models, This paradox occurs alongside dysfunctional or compensatory mechanisms in receptor complex expression. Consequently, while FGF21 levels may have protective effects against T2DM in its early phase, chronic hyperglycemia-induced FGF21 resistance is proposed to be the mechanism explained by the result of our study, where increased serum FGF21 levels initially aim to repair microvascular damage in retinopathy, serving as a counteractive mechanism against metabolic stress and vascular endothelial damage in DR (Frudd et al., 2022; Jin, Xia & Han, 2021; Rusu et al., 2017).

This mechanism occurs similarly in the microvascular protection effect by FGF21. As a regulator of glucose metabolism, FGF21 promotes intrahepatic gluconeogenesis during starvation. However, under hyperglycemic states, FGF21 suppresses glucogenic gene expression, reducing hepatic glucose production and maintaining glucose homeostasis (Erickson & Moreau, 2017; Geng, Lam & Xu, 2020).

It is worth noting that administration of FGF21 has been shown to increase plasma adiponectin concentrations significantly. Adiponectin, an insulin-sensitizing, anti-inflammatory, anti-atherosclerotic, and hepatoprotective factor predominantly produced from adipocytes, contributes to DR ameliorating (Erickson & Moreau, 2017). This finding is further supported by a study by Tomita et al. (2020b, 2019), which demonstrated that administration of (PPAR)δ modulator, known to upregulate FGF21 levels, inhibited pathological angiogenesis in the retina of mouse model by suppressing hypoxia-inducible factor (HIF) and VEGF system Tomita et al. (2020b, 2019). Our result suggests that the observed increase of FGF21 levels in DR and STDR, with a more pronounced effect in STDR, could be explained by two potential mechanisms: (1) a compensatory response to FGF21 resistance as β-cell dysfunction worsens, or (2) an effort to repair microvascular damage and inhibit pathological neovascularization in DR cases. This mechanism aligned with previous findings, which explain the U-shaped relationship between FGF-21 and microvascular complications in T2DM (Jung et al., 2017).

Despite our pooled analysis not identifying any clinical indices as an independent factor of DR incidence, previous studies have highlighted significant associations of obesity (BMI >30), high cholesterol levels, HbA1c, and FGF21 associated with DR incidence (Heidari & Hasanpour, 2021). Another study by Esteghamati et al. (2016) revealed that FGF21 levels, duration of DM, and TG levels significantly predicted DR incidence among T2DM patients. Previous studies have identified age, duration of diabetes, hyperglycemia, hypertension, and hyperlipidemia as known risk factors for DR (Lee et al., 2023; Jin, Xia & Han, 2021). This association between serum FGF21 and DR has been supported by previous research (Heidari & Hasanpour, 2021).

However, it is worth noting that different studies utilized different cut-off values of FGF21 as a biomarker for DR. Jung et al. (2017) used cut-off values of ≤113, 113–214, and ≥214 as cut-off values, with higher values in the latter group reflecting higher risks of DR development. On the other hand, Heidari & Hasanpour (2021) predict the incidence of DR with the optimal cut-off value of >312 pg/mL with sensitivity of 97.80 (92.3–99.7) and specificity of 96.77 (90.9–99.3). Both NPDR and PDR were predicted under the AUC 0.990 model (Heidari & Hasanpour, 2021). In a study by Esteghamati et al. (2016), the clinical cut-off for the pooled samples was 135/5 pg/ml L with a sensitivity of 97.8% and specificity of 75.0%. This study stated that patients with serum FGF21 ≥ 135.5 pg/mL had a 25.86-fold increased risk of T2DR (Esteghamati et al., 2016). The proposed cutoff of FGF21 to predict T2DR, 135.5 pg/mL, is much lower than that reported by Lin et al. (2014). In Lin et al. (2014) study, the mean serum FGF21 levels were 125.9, 326.8, 631.9, and 669.4 pg/ml in controls, T2DM patients without retinopathy, NPT2DR, and PT2DR patients, respectively, which is higher than that calculated for all groups in Esteghamati et al. (2016), study. On the other hand, Mousavi et al. (2017) found the best cut-off values for FGF21 in T2DM at 196 pg/mL, with a sensitivity of 80% and specificity of 47.2%. Hence, the disparities of optimal cut-off values across studies might influence the potential bias within the analysis. Further studies should investigate the optimal cut-off values of FGF21 to validate its clinical utility and enable it as a biomarker in clinical settings.

According to Lin et al. (2014), the proposed independent factors of DR were FGF21, age, diabetes duration, and HDL levels. On the other hand, independent factors for STDR incidence were FGF21, age, diabetes duration, and diastolic blood pressure (Lin et al., 2014) divided their samples into four quartiles with different levels of FGF21, with Q1 being the lowest (FGF21 < 388 pg/mL) and Q4 being the highest (FGF21 ≥ 580 pg/mL). From their pooled analysis, the patients in Q4 had a higher prevalence of DR and STDR (p < 0.05). Serum FGF21 level >478.76 pg/mL suggested the occurrence of DR, and a level >554.69 pg/mL indicated STDR (p < 0.01).

The finding of this study is supported by a study by Jin, Xia & Han (2021), which classified T2DM patients based on their FGF21 serum levels. This found that patients in the highest quartile (Q4) had significantly higher risks of DR and STDR than those in the lowest quartile (Q1), even after adjusting for confounding factors. ROC analysis indicated that serum FGF21 levels above 554.69 pg/mL were associated with an over eight-fold increased risk of STDR (Jin, Xia & Han, 2021). These findings align with our pooled analysis, which demonstrated that higher concentrations of FGF21 predict the severity of DR.

Despite the findings, the mechanisms behind increased serum FGF21 levels in patients with DR remain unclear. Elevated FGF21 in these patients may be a compensatory response to metabolic stress, known as FGF21 resistance (Yang et al., 2012). This resistance, characterized by increased circulating FGF21 and decreased receptor expression, has been associated with a compensatory increase in adiponectin levels in obese individuals, those with insulin resistance, and heart failure patients (Holland et al., 2013; Lin et al., 2013).

FGF21 may target the vascular system, protecting against atherosclerosis by inducing adiponectin to inhibit neointima formation and inflammation and suppressing hepatic cholesterol synthesis to reduce hypercholesterolemia (Rusu et al., 2017; Lin et al., 2015). It also promotes angiotensin II metabolism in adipocytes and renal cells, mitigating hypertension and vessel injury (Lin et al., 2015). Ying et al. (2019) found that FGF21 improved aortic dilation in diabetes mice via oxidative stress suppression and endothelial nitric oxide synthase activation.

The correlation between FGF21 and DR incidence resembles hyperglycemia-associated adiponectin resistance. Increased serum FGF21 may compensate for endothelial dysfunction in retinopathy, with defects in FGF21 expression or activation reducing insulin sensitivity, liver fatty acid oxidation, and triglyceride clearance (Kharitonenkov et al., 2005; Yang et al., 2012; Holland et al., 2013; Rusu et al., 2017). Elevated FGF21 levels in conditions like metabolic syndrome, obesity, insulin resistance, diabetes, and hypertension suggest a response to poor metabolic status (Tan et al., 2023; Gao et al., 2019).

Understanding the complex relationship between FGF21 and diabetes and its complications has paved the way for novel pharmaceutical strategies to overcome T2DM. Tomita et al. (2020a) demonstrated that long-acting FGF21 could reduce retinal vascular leakage in mice with retinal disorders. This finding is further supported by a study by Fu et al. (2017), which revealed that FGF21 administration suppressed ocular neovascularization in mice through adiponectin-mediated pathways. Additionally, FGF21 inhibited pro-inflammatory agents, such as tumor necrosis factor-α (TNF-α), expression but did not alter Vegfa expression in neovascular eyes in mice models. Prior studies have also highlighted the beneficial effects of selective PPARα modulators (SPPARMα), such as fenofibrate, in preventing pathological retinal neovascularization by upregulating liver FGF21 levels (Tomita et al., 2019). These findings suggest that FGF21 could be a therapeutic target for managing pathological vessel growth in DR. Notably, administration of FGF21 has been shown to not only improve the metabolic benefits of insulin sensitivity but also improve lipid profile and obesity, which also reduces the risks of metabolic syndrome (Geng, Lam & Xu, 2020). However, clinical trials investigating FGF21 therapeutic potential in human DR are still limited.

However, the discrepancies between clinical studies, especially in the context of an optimal cut-off value of FGF21 levels in predicting DR incidence and severity, may stem from differences in participant characteristics, such as age, BMI, duration of diabetes, glycemic control, and laboratory methods in FGF21 measurements (Jin, Xia & Han, 2021). Our main finding indicates that elevated serum FGF21 predicts the incidence of DR and STDR among T2DM patients, although the optimal cut-off value remains unclear.

Building on the findings of this study, clinical efforts should be directed toward integrating serum FGF21 level measurement into practice for early detection and management of DR. As the FGF21 measurement is deemed efficient and feasible in community hospitals, our study supports its utilization for eye exams in T2DM patients. Moreover, research should focus on validating optimal cut-off values of FGF21 for DR prediction and exploring the clinical trials in humans for FGF21-based treatments for DR. The significant association between FGF21 levels and DR offers clinicians and researchers insight into a novel pathway for future DR treatment, emphasizing its relevance as a biomarker for monitoring and predicting diabetic complications in T2DM patients.

Conclusions

The serum level of FGF21 is a predictive marker for the incidence of DR in patients with T2DM and demonstrates a positive correlation with the severity of DR in T2DM patients. Thus, FGF21 holds potential as a biomarker for predicting the incidence of DR and determining the prognosis of T2DM. Understanding the link between serum FGF21 levels and DR suggests a pathway for future DR treatment by managing pathological neovascularization via inhibiting pro-inflammatory agents and adiponectin pathways.

Supplemental Information

Supplemental Information 1 PRISMA checklist.

Supplemental Information 2 Details of full search strategy.

Supplemental Information 3 Raw data for the pooled analysis.

Additional Information and Declarations

Competing Interests

Author Contributions

Data Availability

The authors declare that they have no competing interests.

Herni Basir conceived and designed the experiments, performed the experiments, prepared figures and/or tables, authored or reviewed drafts of the article, and approved the final draft.

Annisa Salsabilla Dwi Nugrahani performed the experiments, analyzed the data, prepared figures and/or tables, and approved the final draft.

Andi Makbul Aman conceived and designed the experiments, performed the experiments, authored or reviewed drafts of the article, and approved the final draft.

Syakib Bakri performed the experiments, authored or reviewed drafts of the article, and approved the final draft.

Haerani Rasyid performed the experiments, prepared figures and/or tables, authored or reviewed drafts of the article, and approved the final draft.

Husaini Umar conceived and designed the experiments, prepared figures and/or tables, authored or reviewed drafts of the article, and approved the final draft.

Faridin H. P. conceived and designed the experiments, prepared figures and/or tables, authored or reviewed drafts of the article, and approved the final draft.

Andi Muhammad Ichsan conceived and designed the experiments, prepared figures and/or tables, authored or reviewed drafts of the article, and approved the final draft.

Andi Alfian Zainuddin conceived and designed the experiments, authored or reviewed drafts of the article, and approved the final draft.

The following information was supplied regarding data availability:

This is a systematic review/meta-analysis.

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
