# Peer review of "The association between fibroblast growth factor 21 with diabetes retinopathy among type 2 diabetes mellitus patients: a systematic review, meta-analysis, and meta-regression"

_PeerJ, doi:10.7717/peerj.18308_

## Round 0.1 · original submission · Major Revisions

Dear Authors,

Thank you for submitting your manuscript to our journal. After careful consideration of the feedback provided by the reviewers, I have reached a decision regarding your manuscript.

Reviewer 1 acknowledged the importance and potential impact of your study, highlighting the association between fibroblast growth factor 21 (FGF-21) and diabetic retinopathy (DR) in type 2 diabetes mellitus (T2DM) patients. The reviewer recommended minor revisions, focusing primarily on improving the clarity and grammatical accuracy of the manuscript. They also suggested several specific adjustments, including the need for more detailed explanations of the role of FGF-21 in glucose metabolism, insulin resistance, and DR, as well as the clarification of cutoff values for FGF-21 as a biomarker.

Reviewer 2 provided more extensive feedback and recommended major revisions. While recognizing the value of your research, the reviewer pointed out several areas that require significant improvement. These include enhancing the consistency and clarity of your use of abbreviations, providing more detailed background information on T2DM and DR, and expanding the discussion on the significance of your findings. They also suggested elaborating on the study’s future directions and improving the methodological rigor by properly citing statistical tests and tools.

After reviewing the comments, I agree with Reviewer 2 that your manuscript would benefit from more substantial revisions. Therefore, I am requesting that you address the following points in your revised manuscript:

1. Clarify Abbreviations and Terminology: Ensure consistent use of abbreviations throughout the manuscript. Introduce all abbreviations (e.g., T2DM, DR, FGF-21, VEGF, LDL, HDL) upon first use.
2. Improve Background and Context: Provide a more detailed introduction to T2DM and its relationship with DR. Additionally, clarify the contentious aspects of previous research on FGF-21, as mentioned in your abstract and introduction.
3. Enhance Methodological Transparency: Ensure that all methods, statistical tests, and tools used in your analysis are properly cited. Clarify any ambiguous points in your methods section, such as the databases referred to and the rationale behind certain methodological choices.
4. Expand the Discussion and Conclusion: Further elaborate on the implications of your findings, particularly the relationship between T2DM, DR, and FGF-21. Discuss the potential clinical applications of FGF-21 as a biomarker for DR and explore future research directions based on your findings.
5. Correct Grammatical Issues and Improve Readability: Review the manuscript carefully to address the grammatical issues highlighted by both reviewers. Consider having the manuscript reviewed by a native English speaker to ensure clarity and readability.

Please address these concerns in your revised manuscript and provide a detailed response to the reviewers’ comments, outlining how each point has been addressed.

I look forward to receiving your revised manuscript.

·

Basic reporting

The manuscript submitted 103488v1 explores the association between fibroblast growth factor 21 (FGF-21) with diabetes retinopathy (DR) among type 2 diabetes mellitus (T2DM) patients. Authors illustrate that DR is a leading cause of vision disorders worldwide. FGF-21plays a crucial role in blood sugar regulation and have reported to be correlated with DR incidence and severity and they have concluded that higher level of FGF-21 is associated with higher DR and STDR incidence among T2DM patients. This study supports its use for further eye exams and therapies associated particularly with DR. The association between FGF-21 levels and DR will provide a way to clinicians and researchers to get insight into a novel pathway for future DR research, emphasizing its relevance as a biomarker for monitoring and predicting diabetic complications in type 2 diabetes patient. Clear and unambiguous English is used throughout the manuscript but the manuscript should be carefully revised to correct numerous grammatical issues. Sufficient data is provided to support the findings.

Experimental design

The study is well designed and the findings are interesting and have scientific values. Experimental design is impressive.

Material and Methods
1. Line 154- klotho association with diabetes and retinopathy, needs to be define in introduction.
2. Line 174- authors have not included TG levels as independent variable also dependent variables are not mentioned.
3. Line 183- I2, 2 should be written as subscript as it is creating confusion.
4. Line 193 pooled ORS, What is the significance of doing sensitivity analysis in present review)

Validity of the findings

Results

1. Line 237, what does it mean by the present study demonstrate a significant inverse association between FGF-21 levels and retinopathy diabetes incidence. It is suggested to give more explanation.

Discussion

1. Line 308-313 in discussion is in repetition to line 84 to 88 in introduction, again 313-315 is in repetition.
2. Clearly explain the role of FGF-21 in glucose metabolism and insulin resistance and how increase FGF-21 can affect glucose metabolism. The relation is not clear here, how increase hyperglycemia is related to increase FGF-21 levels and how increase FGF-21 can impair glucose metabolism. The role of FGF-21 with hyperglycemia and DR need more clear consideration.
3. Line 317-325 requires further explanation.
4. FGF-21 leads towards increase gluconeogenesis, so how can FGF-21 aim to repair microvascular damage in retinopathy.
5. More clarification regarding the cutoff values of FGF-21 is required to use it as a biomarker. Are the authors agreed with Jin et al., cutoff value 554.69 pg/ml.

Conclusion
In conclusions, authors have suggested that FGF-21 levels can be used to explore new pathway for future for DR treatment. How kindly explain.

Additional comments

The manuscript should be carefully revised to correct numerous grammatical issues. Additionally, the authors should address the specific issues listed below:
Abstract:
1. Line 25, Diabetic retinopathy (DR) is a leading cause of vision worldwide. It is a mistake, correct this sentence, word problem/ disorder is missing here.
2. Various abbreviation are used without defining them first such as T2DM, STDR and LDL. It is suggested to define them first and then use abbreviations throughout the manuscript.

Introduction:
1. Use of abbreviation is inconsistent, at some places the abbreviation of type 2 diabetes mellitus is used as T2DM and at some places it is written as Type 2 DM (Line 56). Again line 68 and 71, full form of diabetic retinopathy is used instead of abbreviation though it is already defined. Line no. 87 VEGF is not defined here. More consideration is required in this regard.
2. Line 59-61, it is suggested to either break the sentence or rephrase it for better understanding.
3. It would be worthwhile to give a brief detail regarding how oxidative stress can lead to diabetic retinopathy.
4. How proliferative and non-proliferative diabetic retinopathy are different from each other.
5. Line no. 88-90, how this is connected with T2DM, what is the purpose of adding this sentence.
6. Text is not justified throughout the manuscript.
7. It would be beneficial to explore the role of different other pathways associating T2DM and DR.

Reviewer 2 ·

Basic reporting

This study analyzes the association of FGF-21 with DR in T2DM using meta-analysis and meta-regression. However, several parts need to be improved, as stated below.

Experimental design

The methods described can be improved, as commented below.

Validity of the findings

The conclusions can also be improved and the comments are listed below.

Additional comments

1. In the abstract, I think it is better to explain on contentious briefly. Contentious in terms of what aspect?
2. Abstract: A brief introduction of the relationships between DR and Type 2 Diabetes Mellitus may be added.
3. Abstract: T2DM, LDL and STDR stand for?
4. The ultimate contribution should be added in the conclusion of abstract.
5. There is no direct introduction to T2DM. Only introduction to DM, which I think T2DM is one of the main issues in this study, which should be further introduced.
6. Line 87: VEGF stands for?
7. Be consistent in using abbreviations or full names. For example, T2DM, type 2 DM, DR, diabetic retinopathy, FGF21, FGF-21, fibroblast growth factor 21, etc.
8. There is no continuation between sentences in the second paragraph of the Introduction.
9. Line 99 – 100: What do you mean about the controversial conclusion? I think you should write a very specific on the controversial conclusion regarding FGF21.
10. The significance of this study should be added at the end of the Introduction.
11. Line 116 – 117: Both databases refer to which databases?
12. Line 155, diabetes and retinopathy, is this correct?
13. The abbreviations used need to be introduced, e.g., LDL, HDL, etc.
14.The methods, statistical tests and tools used need to be properly cited.
15. I2, I2 and I2 are the same or not?
16. Line 210: Can you elaborate on the “manually from previous reviews”?
17. The future direction/ study from this analysis should be further elaborated.
18. The relationship between T2DM, DR, and STDR should be further explained.

---

## Round 0.2 · accepted · Accept

The authors addressed the reviewers' concerns and substantially improved the manuscript's content. So, based on my assessment as an editor, no further revisions are required and the manuscript can be accepted.

·

Basic reporting

Majority of the changes suggested is initial review with respect to grammatical errors are clearly incorporated in the present manuscript.
Kindly consider line 568-569 in conclusion again.

Experimental design

No comment

Validity of the findings

No Comment

Additional comments

Authors have clearly included all the suggestions.